# LmjF.22.0810 from *Leishmania major* Modulates the Th2-Type Immune Response and Is Involved in Leishmaniasis Outcome

**DOI:** 10.3390/biomedicines8110452

**Published:** 2020-10-26

**Authors:** Andrés Vacas, Celia Fernández-Rubio, Esther Larrea, José Peña-Guerrero, Paul A. Nguewa

**Affiliations:** 1Department of Microbiology and Parasitology, ISTUN Institute of Tropical Health, IdiSNA (Navarra Institute for Health Research), University of Navarra, E-31008 Pamplona, Navarra, Spain; avacas@alumni.unav.es (A.V.); cfdezrubio@unav.es (C.F.-R.); jpena.1@alumni.unav.es (J.P.-G.); 2ISTUN Institute of Tropical Health, Navarra Institute for Health Research (IdiSNA), University of Navarra, E-31008 Pamplona, Navarra, Spain; elarrea@unav.es

**Keywords:** *Leishmania*, NTD, LmjF.22.0810, immune response, leishmaniasis outcome

## Abstract

A novel serine/threonine protein kinase, LmjF.22.0810, was recently described in *Leishmania major*. After generating an *L. major* cell line overexpressing LmjF.22.0810 (named LmJ3OE), the ability of this novel protein to modulate the Th2-type immune response was analyzed. Our results suggest that the protein kinase LmjF.22.0810 might be involved in leishmaniasis outcomes. Indeed, our study outlined the LmJ3OE parasites infectivity in vitro and in vivo. Transgenic parasites displayed lower phagocytosis rates in vitro, and their promastigote forms exhibited lower expression levels of virulence factors compared to their counterparts in control parasites. In addition, LmJ3OE parasites developed significantly smaller footpad swelling in susceptible BALB/c mice. Hematoxylin–eosin staining allowed the observation of a lower inflammatory infiltrate in the footpad from LmJ3OE-infected mice compared to animals inoculated with control parasites. Gene expression of Th2-associated cytokines and effectors revealed a dramatically lower induction in interleukin *(IL)-4*, *IL-10*, and *arginase 1 (ARG1)* mRNA levels at the beginning of the swelling; no expression change was found in Th1-associated cytokines except for *IL-12*. Accordingly, such results were validated by immunohistochemistry studies, illustrating a weaker expression of ARG1 and a similar induction for inducible NO synthase (iNOS) in footpads from LmJ3OE-infected mice compared to control *L. major* infected animals. Furthermore, the parasite burden was lower in footpads from LmJ3OE-infected mice. Our analysis indicated that such significant smaller footpad swellings might be due to an impairment of the Th2 immune response that subsequently benefits Th1 prevalence. Altogether, these studies depict LmjF.22.0810 as a potential modulator of host immune responses to *Leishmania*. Finally, this promising target might be involved in the modulation of infection outcome.

## 1. Introduction

Obligate intracellular pathogens from the genus *Leishmania* kill more than 20,000 people each year. The disease burden attributable to leishmaniasis is close to 2.4 million disability-adjusted life years [1]. The control of leishmaniasis is mainly dependent on chemotherapy [2]. However, current treatment drawbacks and the occurrence of drug-resistant parasites suggest a need to control this disease by applying vaccination or by modulating immune responses.

Leishmanization (LZ) or inoculation of virulent *Leishmania* parasites is the oldest form of vaccination against this disease. It has been largely and successfully implemented in many countries [3]. Additionally, it is the only kind of prophylaxis that has proven efficacy in humans [4]. Despite LZ morbidities, a live vaccine mixture of virulent *L. major* and killed parasites is still used in endemic areas like Uzbekistan [3]. Even though LZ is highly effective, its weaknesses derive from the report of ulcerating primary lesions and some few non-healing lesions [5].

Some novel strategies include the use of genetically modified *Leishmania*, unable to produce the clinical manifestation and capable of altering infection outcomes. The description of factors that regulate and mediate infection-induced resistance may be crucial for designing effective strategies able to modulate leishmaniasis outcome.

During their intracellular stage, parasites infect hematopoietic cells of the monocyte/macrophage lineage through phagocytosis [6]. *Leishmania* has evolved to evade the cytokine-inducible macrophage functions necessary for the development of an effective immune response [6]. As a result, an unbalance in T helper (Th) cells towards the disease-promoting anti-inflammatory subset is produced [7,8]. Furthermore, this augmented Th2 response fosters arginase (ARG) activity in infected macrophages. Concurrently, it suppresses nitric oxide (NO) production while enhancing polyamines production, and consequently facilitates parasite survival [9]. The hallmark of *Leishmania* persistence is their survival ability within the phagolysosome. The induction and suppression of cytokines are some of the mechanisms that *Leishmania* has developed to evade the macrophage antimicrobial machinery [6]. Anti-inflammatory cytokines, such as interleukin IL-10, are induced by parasites in infected macrophages and monocytes [10], promoting their persistence in resistant mice and their multiplication in susceptible mice [11]. IL-10 mediates these outcomes by inhibiting macrophages’ ability to kill parasites by reducing their production of NO and pro-inflammatory cytokines (IL-1, IL-12, and tumor necrosis factor (TNF)) [6]. Transforming growth factor-beta (TGF-β), another cytokine with an anti-inflammatory role, has also been found to be induced by *Leishmania* [12]. It promotes parasites survival by impeding the synthesis of interferon IFN-γ by natural killer (NK) cells, as well as by delaying inducible NO synthase (iNOS) expression in macrophages [6,12]. On the contrary, the pro-inflammatory cytokine IL-12 plays a critical role in the activation of T-lymphocytes and the subsequent secretion of IFN-γ [6]. Thus, IL-12 drives the differentiation and proliferation of Th1 cells via IFN-γ that then leads to the triggering of macrophages and the production of microbicidal molecules [11]. As a response to the beneficial role of IL-12, *Leishmania* has developed the ability to impede its output, regulating the host’s mTOR signaling pathway through the zinc-metalloprotease GP63 [10,13,14]. Additionally, GP63 is also able to cleave the complement protein C3b into C3bi that serves as an opsonin, facilitating parasite uptake through complement receptor 3 (CR3) [15]. Interestingly, the attachment via CR3 facilitates parasites’ “silent entry” into macrophages, through regulating the production of IL-12 [16,17].

We have generated an *L. major* cell line overexpressing the conserved and putatively essential protein kinase LmjF.22.0810 (LmJ3OE). Previous results have shown the implication of this phosphotransferase in the development of resistance to aminoglycosides and susceptibility against amphotericin B and miltefosine [18]. The occurrence and development of resistant and susceptible *Leishmania* strains have previously been associated with parasites harboring modified infectivity profiles [19,20,21,22]. Hence, in the present study, we outlined LmJ3OE infectivity in vitro and in vivo. We found that the increased expression of LmjF.22.0810 not only alters the expression of some known virulence factors, but may also attenuate parasite infections in BALB/c mice by impairing the Th2 immune response. These studies depict LmjF.22.0810 as a new putative target for studying the immunological response to *Leishmania*, and might allow the modulation of leishmaniasis outcomes.

## 2. Materials and Methods

### 2.1. Parasites and Animals

*L. major* (Lv39c5) promastigotes were grown in Schneider’s medium (Gibco Laboratories, Grand Island, NE, USA) supplemented with 10% (*v/v*) heat-inactivated fetal bovine serum (FBS; Gibco Laboratories) and an antibiotic cocktail (50 U/mL penicillin, 50 µg/mL streptomycin; Sigma, St. Louis, MO, USA) at 26 °C.

Infection assays were performed with metacyclic *L. major* promastigotes isolated by the peanut agglutinin (PNA) method [23].

Female BALB/c mice were purchased from Harlan Interfauna Ibérica S.A. (Barcelona, Spain). The study was performed according to ethical standards approved by the Animal Ethics Committee of the University of Navarra, following the European legislation on animal experiments. All the procedures involving animals were approved by the Animal Care Ethics Commission of the University of Navarra (approval number: E5-16(068-15E1) 25 February 2016).

### 2.2. Generation of the Transgenic L. major Strains

For gene overexpression, the coding DNA sequence corresponding to LmjF.22.0810 was amplified by polymerase chain reaction (PCR) from *L. major* genomic DNA as the template, which was extracted following the protocol, described by Medina-Acosta et al. [24]. The primers 5′-AACCCGGGAGTATGGGGCGAGTCGGCGACTAC-3′ and 5′-TACCCGGGCTAAACGTCTCCGCAGTATCC-3′ were used as forward and reverse, respectively, for PCR, and the amplified product was ligated into a pCR 2.1-TOPO (ThermoFisher Scientific, Rockville, MD, USA) cloning vector following manufacturer’s protocol. Then, the plasmid was digested with SmaI (Clontech, Mountain View, CA, USA) to insert LmjF.22.0810 into the *Leishmania* expression plasmid pXG-Hyg. LmjF.22.0810 sequence and orientation within the constructed vector pXG-Hyg-LmjF.22.0810 was assessed by DNA sequencing and PCR, respectively. A total number of 10^8^ log-phase *L. major* parasites were used to transfect the plasmid by electroporation following the method described by Cruz et al. [25]. The overexpressing (LmJ3OE) and the mock control (LmMC) parasites were transfected with pXG-Hyg-LmjF.22.0810 or pXG-Hyg plasmids, respectively. Recombinant colonies were isolated from M199 agar plates supplemented with 100 μg/mL of hygromycin B Gold (InvivoGen Europe, Toulouse, France). LmjF.22.0810 overexpression level was checked in isolated colonies by the quantitative real-time PCR (qPCR) method, following the instructions further described in Section 2.6.

### 2.3. In Vitro Peritoneal Exudate Macrophage (PEM) Infections

Peritoneal exudate macrophages (PEMs) from BALB/c mice were obtained as previously described [26]. Prior to infection with parasites, PEMs were seeded into LabTek plates (50,000 cells/well) containing RPMI 1640 medium (Gibco Laboratories) plus 10% FBS, for 24 h. Macrophages were infected by LmJ3OE or *Leishmania major* mock control peanut agglutinin-negative (LmMC PNA(-)) metacyclic promastigotes parasites, at a ratio of 10:1 (parasites/macrophages), and incubated at 37 °C in a 5% CO_2_ atmosphere. Plates were washed with phosphate-buffered saline (PBS; Gibco Laboratories) and fixed with methanol at 6 h post-infection. All fixed parasites were stained with Giemsa, and the number of amastigotes per macrophage was examined by light microscopy (100 infected cells were examined in each well). Experiments were independently repeated three times in triplicates.

### 2.4. L. major Amastigote RNA Retrieval

*L. major* amastigotes were obtained as follows: bone marrow-derived macrophages (BMDMs) were isolated from female BALB/c mice, as previously described [27]. Then, the BMDMs were plated with a density of 10^6^ cells per well in 24-well plates. After 24 h of incubation, macrophages were infected with LmJ3OE or LmMC PNA(-) metacyclic promastigotes parasites, at a ratio of 10:1 (parasites/macrophages). At 24 h post-infection, non-phagocytosed parasites were washed off with PBS, and infected BMDMs were further incubated for 96 h. At this time, BMDMs containing LmJ3OE or LmMC amastigotes were collected with 200 µl of RNA lysis buffer (Promega, Madison, WI, USA) and stored at −80 °C until RNA extraction.

### 2.5. L. major In Vivo Infections

Fifty-two BALB/c mice were divided into three groups: 22 were infected with LmJ3OE parasites, 22 were infected with LmMC parasites, and 8 were maintained uninfected. The infected animals were subcutaneously inoculated using an insulin syringe into the right hind footpad, three times in successive weeks, with PBS containing 500 PNA(-) metacyclic promastigotes. The uninfected animals were similarly inoculated with PBS solution. Footpad swelling was measured weekly with a digital caliper, and the “net swelling” was determined as the difference between the infected and non-infected footpad.

The detection of a slight swelling was possible in some of the animals at day 23 after the last parasite inoculation. At this time point, 28 animals were euthanized (eight uninfected, ten infected with LmMC, and ten infected with LmJ3OE-1). These euthanized animals were grouped for tissue-RNA extraction (uninfected *n* = 4; LmMC infected *n* = 5; LmJ3OE-1 infected *n* = 5) and immunohistochemistry studies (uninfected *n* = 4; LmMC infected *n* = 5; LmJ3OE-1 infected *n* = 5). The remaining animals were euthanized at day 44 after the last inoculation, and were grouped for tissue-RNA extraction (LmMC infected *n* = 6; LmJ3OE-1 infected *n* = 6) and immunohistochemistry studies (LmMC infected *n* = 6; LmJ3OE-1 infected *n* = 6). For RNA analysis, footpads were collected in RNAlater solution (Invitrogen, Lithuania), and for immunohistochemistry studies, footpads were collected in 4% (*w/v*) formaldehyde (PanReac, Spain) and subsequently formalin-fixed. Spleen and lymph node tissues were rapidly frozen in liquid nitrogen and stored at −80 °C for further analyses.

### 2.6. RNA Extraction and Gene Expression Analysis

Total RNA from *L. major* promastigotes and amastigotes was extracted using the automated Maxwell system, following the manufacturer’s protocol (Promega, Madison, WI, USA). RNA from mice tissue samples was extracted following TRI reagent manufacturer’s protocol (Sigma, St. Louis, MO, USA). After RNA extraction, samples were treated with an Ambion DNA-free Kit (Invitrogen, Carlsbad, CA, USA) for 1 h before “Stop Buffer” solution was added, and RNA was then quantified. One µg of RNA from each sample was used for retrotranscription with M-MLV reverse transcriptase (Invitrogen), following the protocol of the manufacturer. Then qPCR was conducted in 96-well plates using an Applied Biosystems 7500 Real-Time PCR machine (Applied Biosystems, Foster City, CA, USA), with SYBR green PCR master mix (Applied Biosystems), according to manufacturer’s instructions. *Leishmania* and mouse primers used for qPCR are summarized in Table 1 and Table 2, respectively. *Glyceraldehyde-3-phosphate dehydrogenase* (*GAPDH*) (Table 1) [28] was used as housekeeping gene to normalize *L. major* gene expression. For mouse genes, the *β-actin* reference gene was used to normalize the expression (Table 2) [29]. The amount of each transcript was expressed by the formula 2*^ct^*^(GAPDH or actin)^
^−^
*^ct^*^(gene)^, with *ct* being the point (PCR cycle) at which the fluorescence rises appreciably above the background fluorescence.

### 2.7. Leishmania major Quantification

The quantification of *L. major* burden in different tissues from mice was performed measuring mRNA levels of 18S ribosomal gene from *Leishmania* spp. (Lm18S) by reverse transcription, followed by real-time PCR as previously described, using specific primers (see Table 1) [27].

### 2.8. Histological Analysis and Immunohistochemistry Studies

For immunohistochemistry studies, footpads were formalin-fixed, decalcified in Osteosoft-solution (Merck Millipore, Burlington, MA, USA; 1017281000) for 72 h, paraffin-embedded, and cut into 3 µm thick sections. Some sections were stained with hematoxylin and eosin (HE). Immunohistochemistry technique was applied using the following primary antibodies: rat anti-mouse NIMP-R14 (1:10,000; Abcam, Cambridge, UK; ab2557), rabbit-anti-mouse CD4 (1:1000; Abcam, ab183685), rabbit-anti-mouse CD8 (1:400; Cell Signaling, Danvers, MA, USA; 98941), rabbit anti-iNOS (1:400; Abcam, ab15323) and rabbit anti-ARG1 (1:4000; Sigma, HPA003595). Antigen retrieval was performed treating the samples with 2 µg/mL proteinase K at 37 °C for 30 min (for NIMP-R14), or heating for 30 min at 95 °C in 0.01 M Tris-1 mM EDTA pH 9 in a Pascal pressure chamber (Dako, Glostrup, DK; S2800) (for CD4, CD8, iNOS, and ARG1). In the case of rat primary antibodies, sections were firstly incubated with rabbit anti-rat (Dako, Glostrup, DK; E0468) secondary antibody. Then, the EnVision system (Dako, Glostrup, DK; K4003) was used in all cases, according to manufacturer instructions. For each assay, digital images were scanned using a digital microscope system (Aperio ScanScope CS2, Leica Biosystems, Nussloch, Germany), and snapshots of higher magnification images were captured using image software (Aperio ImageScope, Leica Biosystems, Wetzlar, DE). Finally, the percentage area stained in each image was quantified by counting the number of pixels stained above a threshold intensity, and normalizing to the total number of pixels. The software used was Fiji 2.0 [30].

### 2.9. Statistical Analyses

Statistical analyses were executed with GraphPad Prism v7. The analyses were performed using the non-parametric Kruskal–Wallis and Mann–Whitney U tests. All *p* values were two-tailed and considered significant if *p* < 0.05. Data were represented as mean ± SD.

## 3. Results

### 3.1. LmjF.22.0810 Overexpression Conferred to Parasites Lower Infectivity In Vitro

In this paper, after generating *L. major* cells overexpressing the conserved and putatively essential protein kinase LmjF.22.0810, we assessed the rate of infectivity of LmJ3OE parasites in PEM obtained from BALB/c mice. For that purpose, PEMs were infected with LmJ3OE or LmMC PNA(-) metacyclic promastigotes, and 6 h later, the number of infected PEMs and the parasites per cell were determined under a light microscope. Assays were performed with two different strains of LmJ3OE parasites (overexpressing *LmjF.22.0810*) and one of LmMC (as control with normal expression of *LmjF.22.0810*). Figure 1a shows the expression levels of *LmjF.22.0810* in the three *L. major* strains (LmMC, LmJ3OE-1, and LmJ3OE-2). LmJ3OE strains showed significantly higher expression levels of LmjF.22.0810 when compared to control parasites. In fact, LmJ3OE-1 and LmJ3OE-2 exhibited 11.65 ± 3.44- and 18.14 ± 6.72-fold changes, respectively (Figure 1a).

Features as a percentage of viable cells, metacyclogenesis efficiency, and metacyclic-specific cell size remained similar between LmJ3OE and LmMC parasites (data not shown). However, a significant decrease in the percentage of infected PEMs (Figure 1b,d) as well as in the number of amastigotes per infected cell (Figure 1c,d), was observed in both LmjF.22.0810 overexpressing strains with respect to the control strain at 6 h post-infection (Figure 1b–d).

### 3.2. LmJ3OE Parasites Displayed Altered Expression of Genes Implicated in Infectivity

To better understand the mechanism involved in the lower infectivity of *LmjF.22.0810*-overexpressing parasites, we analyzed the expression of several genes implicated in *Leishmania* spp. virulence and drug resistance. Therefore, gene expression levels of *GP63*; *quinonoid–dihydropteridine reductase* (*QDPR*); *small, hydrophilic, endoplasmic reticulum-associated protein* (*SHERP*); and the ATP-binding cassette (ABC) transporter *PRP1* were measured in both metacyclic promastigote and amastigote forms. *GP63*, *QDPR*, and the transporter *PRP1* were downregulated in the promastigote form of LmJ3OE strains with respect to the LmMC strain (Figure 2a), whereas *SHERP* mRNA values did not change (Figure 2a). However, mRNA levels of these genes were similar in LmJ3OE and LmMC amastigotes (Figure 2b).

### 3.3. LmjF.22.0810 Overexpressing Parasites Were Less Virulent In Vivo

To evaluate the virulence of LmJ3OE parasites in vivo, we performed an infectivity assay in BALB/c mice. An implemented methodology allowing the development of a localized infection in the footpad that did not disseminate to internal organs was applied. This assay consisted of three low-dose inoculations of metacyclic parasites (500 parasites), overexpressing or not overexpressing *LmjF.22.0810*, administered subcutaneously once a week in the right hind footpad for three weeks (inoculation period) (Figure 3a). After this time, the footpad swelling was measured until the end of the experiment, 44 days from the last inoculation (lesion development period) (Figure 3a). A slight swelling began to be detected in some animals on day 23 after the last parasite inoculation. As illustrated in Figure 3b–d, LmJ3OE parasites’ infections caused a smaller lesion size compared to the mock control. Such differences were statistically significant from day 30 post-inoculation to the end of the experiment (Figure 3b).

According to these data, hematoxylin–eosin footpad sections staining was also performed. This stain shows the general layout and distribution of cells and provides a general overview of a tissue sample structure. It showed a lower infiltrate area in LmJ3OE-infected mice compared to LmMC infected mice, both 23 and 44 days post-inoculation with *L. major* (Figure 4a). Neutrophils (NIMP-14) were one of the cell types present in the infiltrate area (Figure 4b,d). The study of T lymphocyte (CD8 and CD4) infiltration revealed a smaller area stained in the footpad sections from LmJ3OE-infected mice compared to LmMC-infected mice at day 23 post-inoculation with *L. major* (Figure 4c,e,f). However, at day 44 post-infection with *L. major*, the number of CD8 lymphocytes was significantly higher in LmJ3OE-infected mice compared to LmMC-infected mice, while the number of CD4 lymphocytes was similar between LmJ3OE and LmMC (Figure 4c,e,f).

The expression of *LmjF.22.0810* assessed at these time points confirmed the overexpression of *LmjF.22.0810* in the footpad’s lesions from LmJ3OE-infected mice (data not shown). Additionally, we analyzed the parasite burden in the footpad lesions from both LmJ3OE- and LmMC-infected mice. At day 23 post-inoculation with *Leishmania* strains, the parasite burden was similar in lesions from LmJ3OE-infected compared to LmMC-infected mice (Figure 5a). However, at day 44 post-inoculation, there was a significantly lower parasite burden in mice infected with LmJ3OE parasites compared to mice infected with parasites control (Figure 5b). *Lm18S* mRNA expression was not detected in the spleen or in lymph nodes from inoculated animals.

### 3.4. LmjF.22.0810-Overexpressing Parasites Displayed an Impairment of Th2 Immune Response

As the immune system plays an important role in *Leishmania*-related pathology, we studied the expression of immunomodulatory genes in footpad lesions. This analysis was performed when a slight swelling began to be detected and at the end of the experiment. It is well known that pro-inflammatory (e.g., IFN-γ, TNF-α, IL-1, IL-12) and anti-inflammatory (e.g., IL-4, IL-10, TGF-β) cytokines play different roles in resistance/susceptibility and the immunopathogenesis of leishmaniasis. At day 23 after the last inoculation, when the inflammation began to appear, we observed that LmJ3OE parasite infection led to a significantly lower mRNA induction of *IL1-0, IL-4* and *ARG1*, and an mRNA increase of *IL12p35* compared to LmMC parasites’ infection (Figure 6). No difference was observed in the mRNA expression of *iNOS, TNF-α, TGF-β*, and *IL-1β* in LmJ3OE-infected mice compared to those inoculated with LmMC (Figure 6). Cytokine mRNA values at day 44 post-inoculation showed similar values in footpads from mice infected with LmJ3OE cells and animals inoculated with LmMC parasites (data not shown).

To confirm these results, we performed immunohistochemical analysis of ARG1 and iNOS proteins in footpad sections from LmJ3OE- and LmMC-infected mice at 23 and 44 days post-infection. As observed in Figure 7, at day 23 post-inoculation with *L. major*, the ARG1 staining was significantly lower in footpad sections from LmJ3OE parasite-infected mice compared to controls (footpad sections from mice infected with LmMC parasites) (Figure 7a,c). At day 44 post-infection, the value of ARG1 staining was similar in footpad sections from both the LmJ3OE- and LmMC-infected mice (Figure 7a,c). However, the values of iNOS protein induction were comparable in footpad sections from LmJ3OE- and LmMC-infected mice at both post-infection times, 23 and 44 days.

## 4. Discussion

It is well-known that protein kinases are essential for the proliferation and survival of *Leishmania* species, and might play a crucial role in parasite infectivity and pathogenicity [31,32,33,34]. Previous reports have described the implication of the trypanosomatid protein kinase LmjF.22.0810 in *Leishmania* paromomycin drug resistance [18]. Here, we aimed to determine the implication of LmjF.22.0810 overexpression in *Leismania* infectivity, virulence, and infection outcome. Interestingly, we observed a lower infection rate in vitro for LmJ3OE parasites compared to LmMC cells. Such decreased infection level prompted the analysis of *Leishmania* genes involved in virulence or drug resistance, such as the zinc–metalloprotease *GP63* [35,36], *SHERP* [37], *QDPR* [38], and the ABC transporter *PRP1* [39,40]. Expression levels of *GP63*, *QDPR*, and *PRP1* were dramatically downregulated in metacyclic forms of LmJ3OE parasites. The surface protease GP63 is known as the major virulence factor found in the glycocalyx of *Leishmania* parasites, which greatly influences host cell signaling mechanisms and related functions [41]. Moreover, GP63 has been found to be involved, not only in the cleavage and degradation of various host proteins, such as C-Jun kinase (JNK) and mTOR, but also to have a meaningful impact on disease progression by inhibiting the host’s Th1 immune response [14,42]. In addition, GP63 increases parasites’ resistance against complement-mediated lysis through the conversion of C3b into C3bi, and by reducing the fixation of terminal complement components to *Leishmania* [15]. On the other hand, the ABC transporter PRP1 is known to protect *L. major* parasites against toxic molecules [39,40], whereas QDPR activity has been described as essential in parasite infectious cycle [38]. Thus, in metacyclic forms of LmJ3OE parasites, the diminished expression levels of genes encoding those proteins may allow *Leishmania* infectivity. In contrast, in the amastigote form, gene expression of these virulence factors was similar in LmJ3OE parasites and controls, suggesting that the overexpression of LmjF.22.0810 plays a more important role in the infectivity of the promastigote form than in the intracellular survival of amastigotes.

The evaluation of some biological characteristics of LmJ3OE parasites did not differ from those found in controls. The cell morphology, cell cycle development, promastigote and amastigote growth, and the percentage of metacyclic cells (data not shown) have been analyzed. The corresponding results were very similar to control outcomes. However, our results showed that LmJ3OE metacyclic promastigotes displayed a low infectivity in vitro and smaller lesion size in vivo. During the in vivo assay, the footpad swelling of mice from both groups (LmMC and LmJ3OE) started at week 3, with similar lesions sizes. However, the footpad thickness significantly increased in the control group after seven days. This difference in the footpad swelling was maintained during the rest of the assay, and was noticeable by the end of the experiment (day 44 post-inoculation), with LmJ3OE infected mice displaying significantly less footpad thickness. Accordingly, hematoxylin–eosin staining showed a smaller infiltration area in footpads from LmJ3OE infected mice compared to control footpads, with neutrophils being one of the cell types present in the infiltrate. This observation is in line with published data, where it has been described that neutrophils are the first cells to massively arrive at the site of *L. major* infections [43].

The quantification of the expression of cytokines and effector molecules after footpad swelling revealed, on the one hand, a slight upregulation of Th2 cytokines in LmJ3OE-infected mice (IL-4 and IL-10) compared to uninfected controls. However, such increases were much lower than those observed in LmMC-inoculated animals. It is well-known that even though a Th2 response commonly prevents tissue destruction, it also promotes the infection by intracellular pathogens, such as *Leishmania* [44], and that mouse strains like BALB/c generally develop a progressive disease caused by early Th2 responses [43,45]. On the other hand, the expression analyses from the LmJ3OE-infected mice also show that except for *IL12p35*, the levels of the evaluated Th1 molecules (TNF-α, IL-1β, and iNOS) remained unchanged when compared to LmMC-infected animals. Therefore, the immune response balance seems to favor Th1 prevalence. Such polarization of immune responses towards Th1 prevalence during cutaneous leishmaniasis in BALB/c mice, has been reported to mediate resistance to infections and to promote wound healing [43,46,47]. The production of IL-4 during the initial phase of infection is sufficient to disrupt the Th1/Th2 balance towards a progressive disease in BALB/c mice, where Th2 cytokines are dominant [45]. Furthermore, the crucial role of IL-10 in *Leishmania* disease progression had been demonstrated [48], as has the detrimental role of neutrophils in the development of immunopathology during cutaneous leishmaniasis in the absence of IL-10 [49,50]. Interestingly, LmJ3OE infections displayed a significantly lower expression of IL-4 and IL-10. Furthermore, according to other experimental *Leishmania* infections with similar outcomes [7,12,46,51], the initial increase of IL-12, as well as the reduction of IL-4 and IL-10, may be responsible for the observed controlled swelling in LmJ3OE-infected mice during the assay. On the other hand, it has been reported that CD8 T cells contribute to the induction of protection against *L. major* [52]. Thus, the observed higher number of CD8 T cells at day 44 in the LmJ3OE-infected footpads compared to the control may contribute to protection.

Footpad swelling reflects not only local inflammation, but also parasite replication. Therefore, the significantly lower parasite load in LmJ3OE-infected mice, even when the footpad thickness was large, was an unusual finding. This finding was in agreement with the lower amount of ARG1 found in LmJ3OE-infected mice, since high ARG1 activity had been associated with increased parasite load in *L. major*-infected BALB/c mice [53,54].

The footpad swelling of the control animals did not allow us to extend the experiment after week 6. In LmJ3OE-infected mice, the clearance was not achieved by the end of the experiment. A similar result had been described during the evaluation of centrin gene-deleted *L. donovani* as a vaccine [55]. In fact, parasites were detectable in the viscera by week 5, and at least 12 weeks were needed before the parasite clearance was achieved in the spleen and liver from infected BALB/c mice [55].

## 5. Conclusions

In conclusion, we show that LmjF.22.0810 overexpression might modulate the host immune response and can be involved in leishmaniasis outcome. We observed that the uptake of LmjF.22.0810-overexpressing parasites seems to negatively regulate the expansion of Th2 cytokines in BALB/c mice. This aids host protection against cutaneous leishmaniasis development by promoting the dominance of Th1 cytokines. The disclosed findings provide the first report to understand how LmjF.22.0810 mediates immune modulation of host–parasite interactions. Additionally, our work describes LmjF.22.0810, a conserved trypanosomatid kinase, as a new and promising target for studying the immune response produced by *Leishmania* infections.

## Figures and Tables

**Figure 1 biomedicines-08-00452-f001:**
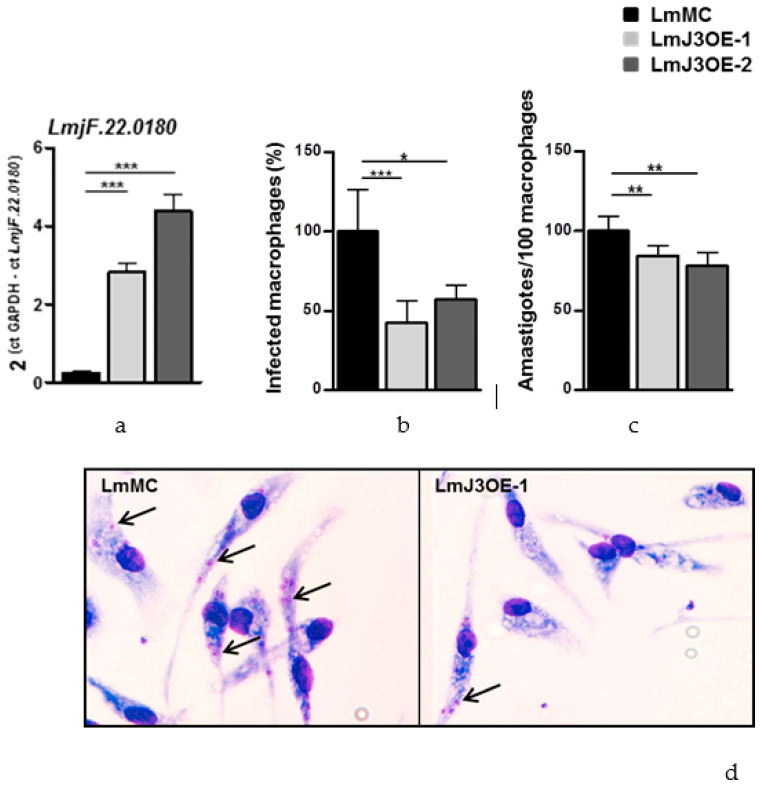
Infectivity rate of LmjF.22.0810 overexpressing and control parasites in peritoneal macrophages from BALB/c mice. (**a**) *Leishmania major LmjF.22.0810* expression of recombinant colonies was checked by quantitative real-time PCR (qPCR) and represented. Assays were performed with two different LmjF.22.0810-overexpressing *Leishmania major* strains (LmJ3OE-1 and LmJ3OE-2), compared to *Leishmania major* control (LmMC) parasites. (**b**) The percentage of infected peritoneal exudate macrophages and (**c**) the number of amastigotes per 100 infected cells was determined by microscopy counting after Giemsa staining at 6 h post-infection. (**d**) The observed changes in the infectivity rate throughout the in vitro infection caused by LmJ3OE and LmMC parasites were graphed for one representative experiment. Solid black arrows indicate Giemsa-stained amastigotes. Data represented the means (±SD) from the triplicates of at least three independent experiments (* *p* < 0.05, ** *p* < 0.01, *** *p* < 0.001).

**Figure 2 biomedicines-08-00452-f002:**
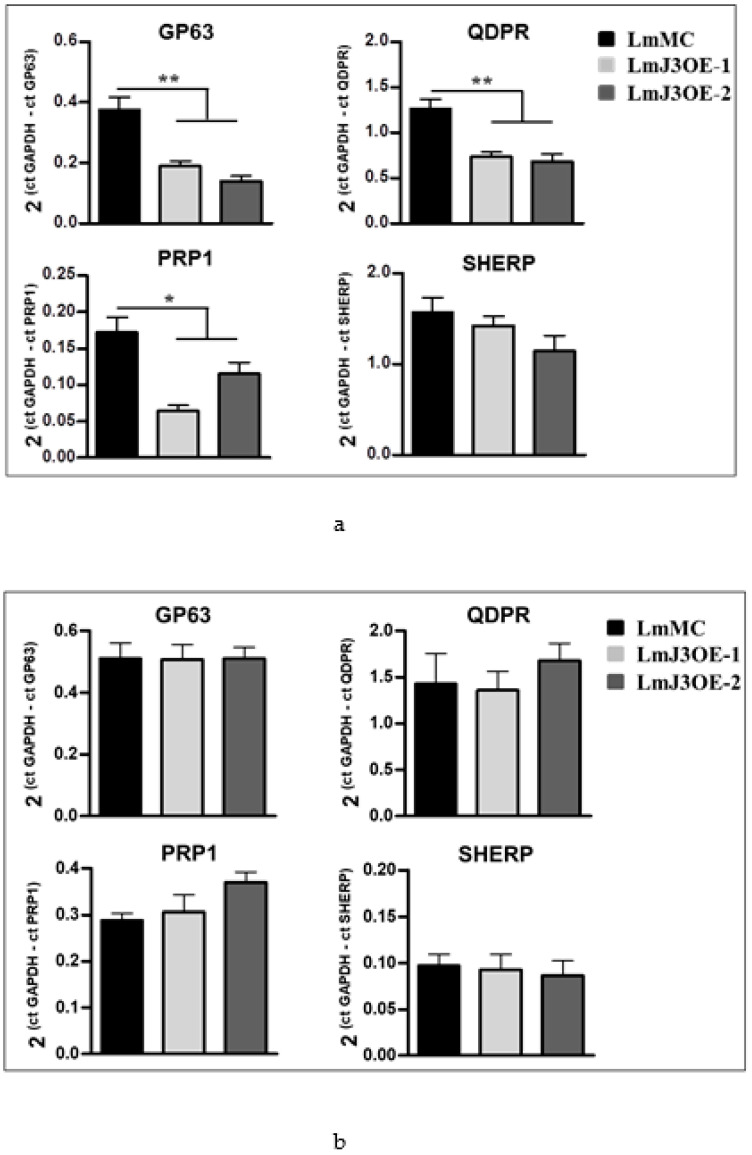
Expression profile of genes involved in the infectivity of LmjF.22.0810-overexpressing parasites compared to control. The mRNA expression levels of the zinc–metalloprotease *GP63*; *quinonoid–dihydropteridine reductase* (*QDPR*); the ATP-binding cassette transporter *PRP1*; and the *small, hydrophilic, endoplasmic reticulum-associated protein* (*SHERP*) were determined by qPCR in the (**a**) metacyclic promastigote and (**b**) amastigote forms of the three parasite strains, overexpressing (LmJ3OE-1 and LmJ3OE-2) or not overexpressing (LmMC) LmjF.22.0810 kinase. Bars represent the means (±SD) from three independent experiments (* *p* < 0.05, ** *p* < 0.01).

**Figure 3 biomedicines-08-00452-f003:**
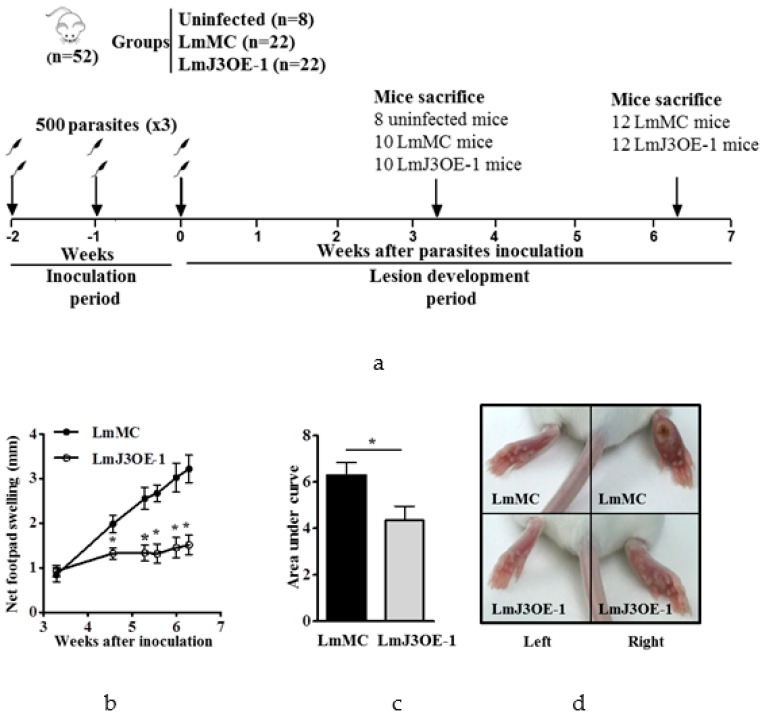
Virulence shown by LmjF.22.0810-overexpressing parasites in BALB/c mice. (**a**) Schematic representation of the experimental setting. During the inoculation period, animals were subcutaneously infected in the right footpad with 500 metacyclic parasites, overexpressing (LmJ3OE-1) or not overexpressing (LmMC) LmjF.22.0810. Three cycles of inoculation were carried out once a week. After the last inoculation, the swelling was measured weekly until the end of the experiment (lesion development period). The “*black arrows*” indicate the three cycles of inoculation (1 per week) and the two time points when mice were euthanized (days 23 and 44 after the last inoculation). (**b**) Net footpad swelling corresponds to the difference between the infected and non-infected footpad, measured weekly until 44 days after last parasite inoculation (both LmMC and LmJ3OE-1 strains). (**c**) Area under the curve for net footpad swelling from mice inoculated with LmMC or LmJ3OE-1 parasites. (**d**) Representative images of inoculated (right) and non-inoculated (left) footpads of mice infected with parasites overexpressing (LmJ3OE-1) or not overexpressing (LmMC) LmjF.22.0810. Bars represent the means (±SD) (* *p* < 0.05).

**Figure 4 biomedicines-08-00452-f004:**
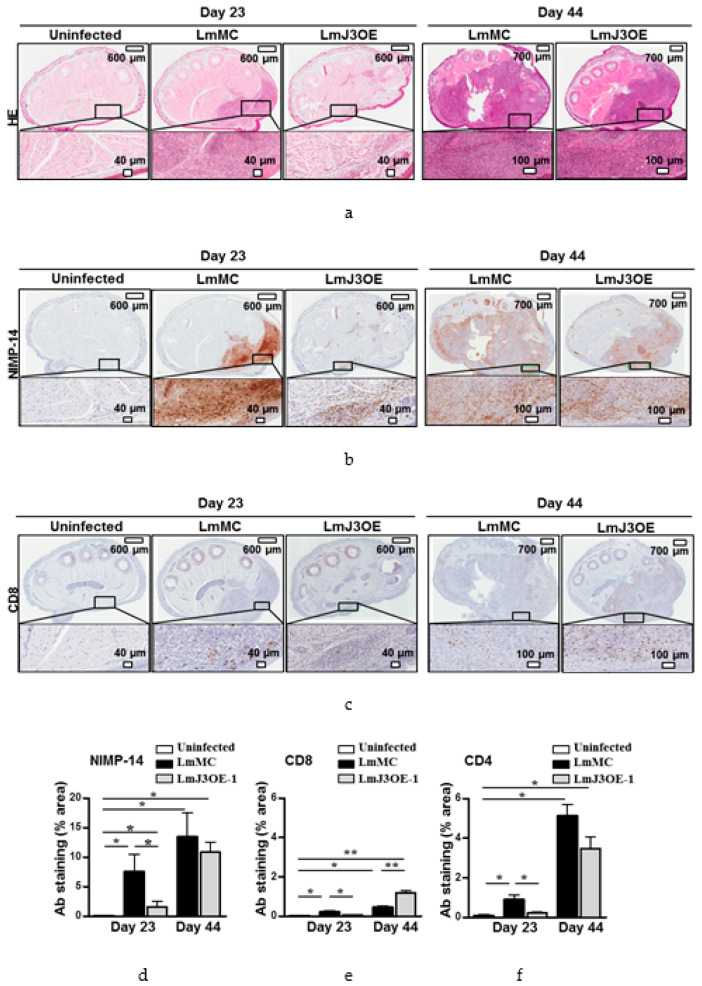
Histopathology and immunohistochemistry results of the footpad sections. Twenty-three and 44 days after the inoculation period, animals were sacrificed in order to study footpad inflammation by histology staining techniques. (**a**) Hematoxylin–eosin and (**b**) immunohistochemical staining for neutrophils (NIMP-14) and (**c**) CD8 T-cells of footpad sections of *L. major*-overexpressing LmjF.22.0810 (LmJ3OE-1)-infected mice compared to *L. major* control (LmMC)-infected mice, both 23 and 44 days after the last inoculation. (**d**) Graphical representation of the footpad section area stained with NIMP-14 (neutrophils), as well as (**e**) CD8 and (**f**) CD4 T cell antibodies of LmJ3OE-infected mice compared to uninfected and LmMC-infected. Bars represent the means (±SD) (* *p* < 0.05, ** *p* < 0.01).

**Figure 5 biomedicines-08-00452-f005:**
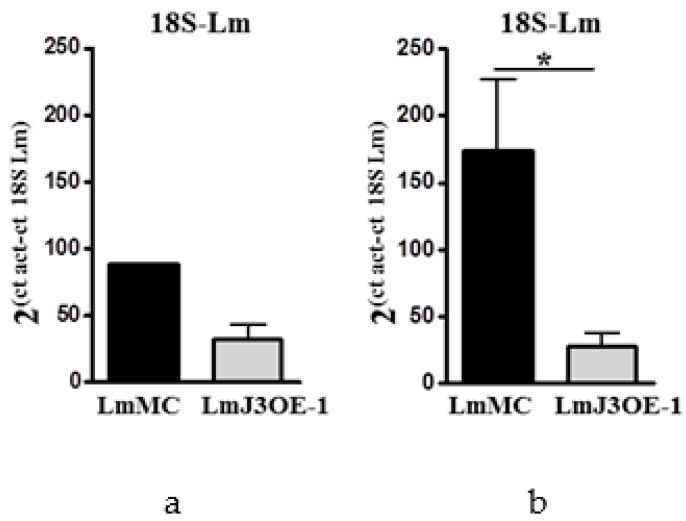
Parasite load levels in the mice footpads inoculated with *LmjF.22.0810*-overexpressing and mock control parasites. Parasite burden was measured by *Lm18S* mRNA expression in the footpad lesions from both *LmjF.22.0810*-overexpressing (LmJ3OE-1) *L. major* and the *L. major* control (LmMC) infected mice at day 23 (**a**) and 44 (**b**) post-inoculation. Bars represent the means (±SD) (* *p* < 0.05).

**Figure 6 biomedicines-08-00452-f006:**
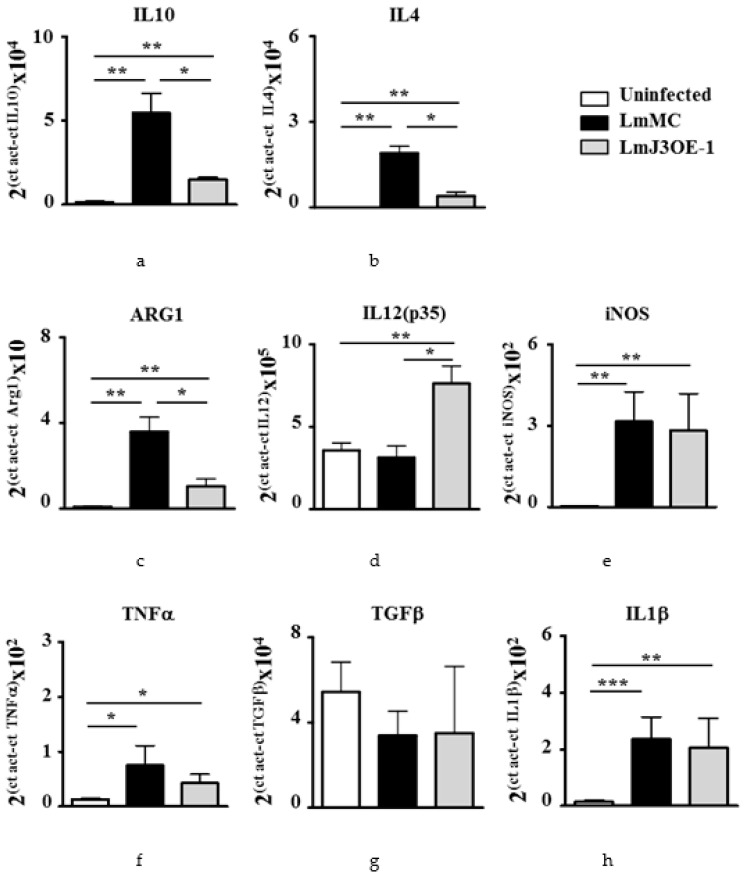
Immune-related gene expression profiles 23 days after inoculation period. mRNA levels from footpad lesions were analyzed by qPCR to study the immunomodulatory role of molecules (**a**) *interleukin (IL)-10*, (**b**) *IL-4*, (**c**) *arginase (ARG)1*, (**d**) *IL-12 (p35)*, (**e**) *inducible NO synthase (iNOS)*, (**f**) *tumor necrosis factor (TNF)-α*, (**g**) *transforming growth factor (TGF)-β*, and (**h**) *IL-1β*. The analysis was performed on day 23 after inoculation with both *L. major* overexpressing *LmjF.22.0810* (LmJ3OE-1) and *L. major* control (LmMC) strains. Bars represent the means (±SD) (* *p* < 0.05, ** *p* < 0.01, *** *p* < 0.001).

**Figure 7 biomedicines-08-00452-f007:**
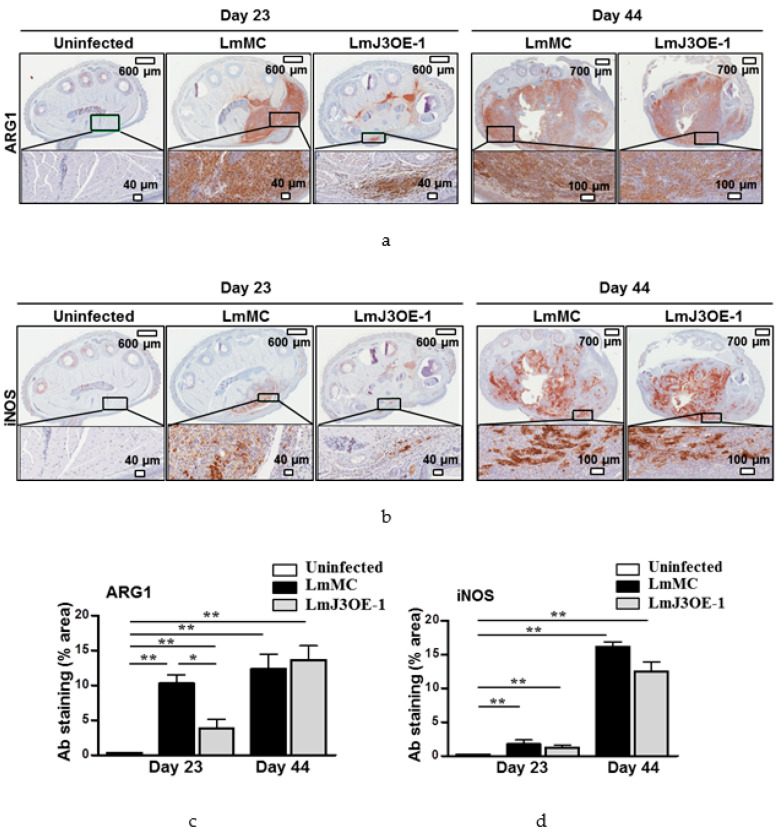
Immunohistochemical analysis of ARG1 and iNOS in footpad sections from LmJ3OE- and LmMC-infected mice. Footpad sections stained with (**a**) ARG1 and (**b**) iNOS antibodies from uninfected mice compared to both *L. major* overexpressing LmjF.22.0810 (LmJ3OE-1) and *L. major* control (LmMC)-infected animals. The differences in the area of footpad sections stained with antibodies of (**c**) ARG1 and (**d**) iNOS from LmJ3OE-infected mice compared to uninfected and LmMC-infected mice are illustrated. Bars represent the means (±SD) (* *p* < 0.05, ** *p* < 0.01).

**Table 1 biomedicines-08-00452-t001:** Primer sequences used for the quantification of *Leishmania* genes.

Gene	Sense Primer (5′→ 3′)	Antisense Primer (5′→ 3′)
*LmjF.22.0810*	cctccacagggaaagcaac	caatgcaccccatcgaccaa
*GP63*	actgcccgtttgttatcgac	ccggcgtacgacttgactat
*SHERP*	gacgctctgcccttcacatac	tctctcagctctcggatcttgtc
*QDPR*	atgaaaaatgtactcctcatcg	ttcaccctgcgtactgaacacat
*PRP1*	ctcatgcgtcagtgcaagtg	aaacaacgggcaaaaagcga
*Lm18S*	ccaaagtgtggagatcgaag	ggccggtaaaggccgaatag
*GAPDH*	accaccatccactcctaca	cgtgctcgggatgatgttta

**Table 2 biomedicines-08-00452-t002:** Primer sequences used for the quantification of mouse genes.

Gene	Sense Primer (5′→ 3′)	Antisense Primer (5′→ 3′)
*IL12p35*	cacgctacctcctctttttg	aggcaactctcgttcttgtg
*IL10*	ggacaacatactgctaaccg	aatcactcttcacctgctcc
*IL1β*	gccaccttttgacagtgatg	taatgggaacgtcacacacc
*ARG1*	tggggaaagccaatgaagag	aggagaaaggacacaggttg
*IL4*	gctattgatgggtctcaacc	tctgtggtgttcttgttgc
*TGFβ*	cggcagctgtacattgac	tcagctgcacttgcaggagc
*TNFα*	cttccagaactccaggcggt	ggtttgctacgacgtggg
*iNOS*	tcctacaccacaccaaactg	aatctctgcctatccgtctc
*β-actin*	cgcgtccacccgcgag	cctggtgcctagggcg

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
