# Peer review of "LmjF.22.0810 from Leishmania major Modulates the Th2-Type Immune Response and Is Involved in Leishmaniasis Outcome"

_biomedicines, 2020, doi:10.3390/biomedicines8110452_

Round 1

Reviewer 1 Report

Minor issues:

line 49: change to “derives”

line 211: what are the differences between the two strains of LmJ30E

line 273: change to “shown”

Major issue: the authors failed to provide the ethical assessment of their research on mice as requested (number of protocol and approving institution)

I am also not sure of the interest of the readers of this journal to the topic described in this article.

Author Response

Dear Reviewer:

Thank you very much for considering our manuscript for resubmission.

Your comments and suggestions made are insightful. We have addressed all these issues and made the appropriate changes in the text; we consider that the manuscript has substantially improved and hope that it is now ready for publication in Biomedicines. Please find our point-by-point response as well as our new manuscript.

Thanks again.

Reviewer 2 Report

This article, that studies the consequences, mainly on infectivity and virulence, of LmjF.22.0810 overexpression, is well written. I just recommend minor modifications before acceptance :

  • Homogeneity is lacking concerning the name of the gene. Indeed, sometimes it is written LmjF.22.0180 but sometimes it is Jean3 or LmJ3. Use only one manner to write the gene name in all the manuscript (text and figures and legends).
  • Sometimes, the name of the gene is not in italics: check all the manuscript. L. 258: in vivo must also be in italics.
  • Some terms need to be better defined the first time. For example, define LmMC PNA(-) line 209 and Lm18S line 319 and IL12p35 line 334 (why p35?). The role of each proteins tested in paragraph 3.2 must be explained (why have they been chosen ?). And the interest/the principle of hematolysin-eosin staining must be explained in the results.
  • l. 80: a space is lacking
  • l. 98: hygromycin B is used only for the transfected strain, not the parent L. major strain (or why is there such a selection pressure in the parent strain?)
  • Fig 2b SHERP: change scale because we cannot see error bars (and the scale is different in the other graphs).
  • Fig 2: what does 'means of one of three experiments performed in sextuplicates' mean ? How can we have a mean with only one experiment ??
  • Fig 3b and c are strictly identical: I think there is a mistake. And change the corresponding letters in the legend.
  • Fig 3: line 276 and 282: 'not' instead of 'no'.
  • l. 286: according to the figure, it seems that it is not significant at day 44 so you cannot say that there is a lower infiltrate area at day 44.
  • l. 287-288 and also l. 409: I don't think you can say that neutrophils are 'the most abundant cells in the infiltarte area' because to which cells have you compared ? You tested only neutrophils, CD4 and CD8 cells...
  • l. 289: it seems, according to the figure 4e and 4f, that there is no increase in LmJO3E compared to the control at day 23 so you cannot say that there is a smaller increase.
  • l. 292-293: if it is not statistically significant, then there is no change: change the sentence. Idem lines 315-316.
  • Fig : bars represent the means from how many experiments ?
  • l. 315-316: revise the sentence
  • I  the results, paragraph 3.4, define at the beginning if the different cytokines tested are anti- or pro-inflammatory.
  • Fig 6d: IL12 is pro-inflammatory so how can we explain the increase when LmjF.22.0810 is overexpressed, overexpression that is associated with less infectivity ?
  • Fig 6: legend: place the letters a, b, c,... in order
  • l. 423: 'to' instead of 'of' ?
  • l. 433: the higher number of CD8 T cells is observed only at day 44, not at day 23 where there is a decrease! Change the sentence.
  • l. 439: not 'in' in front of 'BALB/c mice'.
  • l. 440: it repeats the previous paragraph.

Author Response

Dear Reviewer:

Thank you very much for considering our manuscript for resubmission.

Your comments and suggestions made are insightful. We have addressed all these issues and made the appropriate changes in the text; we consider that the manuscript has substantially improved and hope that it is now ready for publication in Biomedicines. Please find our point-by-point response as well as our new manuscript.

Please see the attachment. Thanks again.

Round 2

Reviewer 1 Report

The authors have specified the comments and corrected/implemented where necessary. The manuscript is now suitable for publication.